# Glycogen Synthase Kinase 3β: A True Foe in Pancreatic Cancer

**DOI:** 10.3390/ijms232214133

**Published:** 2022-11-16

**Authors:** Omer H. M. Elmadbouh, Stephen J. Pandol, Mouad Edderkaoui

**Affiliations:** 1Department of Medicine, Division of Gastroenterology and Hepatology, Samuel Oschin Comprehensive Cancer Center, Cedars-Sinai Medical Center, Los Angeles, CA 90048, USA; 2Department of Biomedical Sciences, Cedars-Sinai Medical Center, Los Angeles, CA 90048, USA

**Keywords:** pancreatic cancer, GSK-3β

## Abstract

Glycogen synthase kinase 3 beta (GSK-3β) is a serine/threonine protein kinase involved in multiple normal and pathological cell functions, including cell signalling and metabolism. GSK-3β is highly expressed in the onset and progression of multiple cancers with strong involvement in the regulation of proliferation, apoptosis, and chemoresistance. Multiple studies showed pro- and anti-cancer roles of GSK-3β creating confusion about the benefit of targeting GSK-3β for treating cancer. In this mini-review, we focus on the role of GSK-3β in pancreatic cancer. We demonstrate that the proposed anti-cancer roles of GSK-3β are not relevant to pancreatic cancer, and we argue why GSK-3β is, indeed, a very promising therapeutic target in pancreatic cancer.

## 1. Introduction

Glycogen synthase kinase 3 beta (GSK-3β) is a serine/threonine protein kinase originally identified as a regulator of glycogen synthase activity. GSK3β is involved in glycogen synthesis and multiple signalling pathways linking the protein to disorders ranging from neurodegenerative diseases to cancers, and involving differentiation, proliferation, motility, and apoptosis. 

Many studies showed pro- and anti-oncogenic roles of GSK-3β in many cancers including in pancreatic ductal adenocarcinoma (PDAC). The detailed mechanisms of the GSK-3β pathways have been previously discussed and can be found in many interesting reviews [1,2,3,4,5,6,7,8]. Here, we do not discuss the detailed pathways regulated by GSK-3β in cancer, but we focus on discussing the relevance to PDAC of the previously proposed anti-cancer and pro-cancer roles of GSK-3β.

## 2. Regulation of GSK-3β Activity

The activity of GSK-3β is mainly regulated by phosphorylation at serine 9 which results in its inhibition. This phosphorylation is regulated by protein kinase A (PKA), Akt kinase, ribosomal protein S6 kinase beta-1 (p70S6K), and p90 ribosomal S6 kinase (p90RSK), which inactivate GSK-3β and inhibit its ability to phosphorylate downstream substrates [9]. Other inactivating phosphorylation reactions of GSK-3β include Threonine 390 and Threonine 43 by p38 mitogen-activated protein kinases (p38 MAPK), and extracellular signal-regulated kinases (ERK) respectively [10,11]. Both p38 MAPK and ERK are important proteins in cancer development and progression. 

Phosphorylation of Tyrosine 216, by autophosphorylation or by Src tyrosine kinase, activates GSK-3β [12,13]. Other regulators of the Tyr 216 phosphorylation/activation include the Ca^2+^-dependent tyrosine kinase Pyk2 and the proto-oncogene tyrosine-protein kinase Fyn [14,15]. Regulators of GSK-3β inhibitory and activating phosphorylations are summarized in Figure 1.

Protein phosphatases 1 and 2A (PP1 and PP2A) act as activating factors by dephosphorylating serine 9 to increase GSK-3β stability and activity. Other regulations of GSK-3β activity include subcellular localization and the formation of protein complexes [16]. 

GSK-3β catalyses the last step of glycogen synthesis and regulates β-catenin, a critical step in the Wnt pathway. These two processes were the first GSK-3β pathways studied in detail. They are also characteristics of aggressive cancers and contribute to the therapeutic hurdle of chemoresistance and high metastatic frequency that continues to maintain survival rates of pancreatic ductal adenocarcinoma (PDAC) patients at low levels. 

The role of GSK-3β in the Wnt pathway, initially, caused GSK-3β to appear as if it has played an anti-cancer role for many years. Indeed, many reviews showed detailed roles of GSK-3β in cancer and suggested it as an anti-cancer protein. The detailed pathways regulated by GSK-3β in cancer in general were discussed in detail in many published reviews [1,2,3,4,5,6,7,8]. In this mini-review, we discuss the suggested anti-cancer roles of GSK-3β and question their relevance to PDAC.

## 3. GSK3-β Expression in Cancer

Overexpression of GSK-3β has been observed in various tumour types, including colon, liver, ovarian, and pancreatic cancers [17,18,19,20]. GSK-3β’s pathological role is marked by upregulation of expression and activity in cancer cells compared to non-neoplastic cells and defined by differential phosphorylation [17]. Increased GSK-3β activity is observed in cancer cells, as well as in metastatic lesions, and is responsible for the inherent chemoresistance observed in pancreatic cancer [17,21]. 

In addition, cell localization of GSK-3β seems to play an important role in cancer development and promotion. For example, nuclear localization of GSK-3β was demonstrated in 70% of human breast carcinomas and 73% of squamous cell head and neck carcinomas, whereas no detectable expression of nuclear GSK-3β was found in benign breast tissue and in benign salivary gland and other benign head and neck tissues [19]. In pancreatic cancer, a study by Ougolkov et al., showed that nuclear localization of GSK-3β is increased with the loss of differentiation of the cancer cells. In fact, they found that PanINs = 1 lesions, which are pre-cancer lesions, had only cytoplasmic localisation of GSK-3β, whereas well-differentiated cancer cells had increased cytoplasmic expression and moderate nuclear localization, and poorly differentiated cancer cells showed strong nuclear localization and little presence in the cytoplasm [22]. Nuclear localization of GSK-3β is associated with a worse outcome and drug resistance in cancer, and the inhibition of GSK-3β activation decreases its nuclear accumulation [22,23]. 

## 4. GSK-3β Proposed Onco-Suppressive Effect in Pancreatic Cancer

One of the first discovered roles of GSK-3β is its regulation of the Wnt/β-catenin signalling pathway. When the Wnt ligand is present, it binds to specific membrane-bound receptors. This binding activates an intracellular signalling cascade resulting in β-catenin stabilization and nuclear localization. In the nucleus, β-catenin associates with members of the TCF/LEF family of transcription factors to regulate the transcription of various Wnt targets, which are responsible for cell growth and motility, and both are important for cancer progression and metastasis. 

GSK-3β phosphorylates β-catenin, triggering its degradation, and consequently reducing β-catenin nuclear accumulation and reversing the pro-cancer Wnt/βcatenin pathway [24]. The β-catenin destruction complex is assembled through the interactions between Adenomatous polyposis coli (APC), AXIN, GSK-3β, and Casein kinase 1 alpha (CK1α). Specifically, APC directly binds AXIN via Ser-Ala-Met-Pro (SAMP) repeat sequences and β-catenin via three 15-amino acid repeats and seven 20-amino acid repeats. AXIN interacts with GSK-3β, CK1α, and β-catenin, in addition to its binding to APC. The complex is stabilized via GSK-3β and CK1α-mediated AXIN phosphorylation. CK1α phosphorylates serine 45 of β-catenin, which primes the subsequent, sequential, phosphorylation of Threonine 41, serines 33 and 37 by GSK-3β. After these phosphorylation events, β-catenin is retained in the complex, thereby promoting its recognition by the Beta-transducin repeats-containing proteins (β-TrCP) for polyubiquitination on Lys-19 [25]. Therefore, in this case, inhibition of GSK-3β will lead to activation of β-catenin and its pro-cancer activated pathway leading to cancer promotion. 

In contrast to this process, recent studies have shown that inhibition of GSK-3β leads to the stabilization of β-catenin and induces PDAC cell death with β-catenin having no effect on PDAC cell survival [26,27]. Importantly, recently published data show that β-catenin partly mediates the killing effects of GSK-3β pharmacological and molecular inhibitors in K-ras-dependent tumors [28]. Pancreatic cancer cells infected with lentivirus containing Cas9 and guide RNA (gRNA) to β-catenin or scrambled were treated with GSK-3β inhibitor SB-732881-H to test whether β-catenin inhibition will rescue PDAC cells from apoptosis induced by the GSK-3β inhibition. The results showed that in scrambled gRNA-infected cells, GSK-3β inhibition upregulated β-catenin, and induced caspase-3 activation and apoptosis. In contrast, a targeted knockout of β-catenin in cells infected with β-catenin gRNA abrogated GSK-3β inhibition induced apoptosis [28]. Taken together, apoptosis induced by GSK-3β inhibition in PDAC tumors is in part mediated by β-catenin. Interestingly, extended overexpression of β-catenin in various cancer cell lines from different origins induced apoptosis [29]. Based on this data, we can state that the GSK-3β/β-catenin pathway is a pro-cancer pathway in PDAC.

In addition to the GSK-3β/β-catenin pathway, multiple studies indicate that GSK-3β-mediated regulation of the Wnt pathway can modulate downstream signalling pathways independently of β-catenin. One major pathway is the inhibition of mTOR by GSK-3β mediated by phosphorylation of Tuberous sclerois complex 2 (TSC2). This pathway is activated by Wnt/GSK-3β independently of β-catenin [30] and is considered another anti-oncogenic effect of GSK-3β in PDAC as mTOR is a major pro-cancer effector in PDAC [31,32]. However, these findings were obtained in healthy cells including fibroblasts, bone marrow, and kidney cells; while data in cancer cells showed that overexpression of GSK-3β activated mTOR complex-1 (mTORC1) leading to activation of the mTOR pathway, indicating that the GSK-3β/mTOR interaction in cancer cells is different from normal cells [33].

In addition to the Wnt pathway, another proposed onco-suppressive effect of GSK-3β is its phosphorylation of the tumour suppressor protein p53, inducing its activation and transcriptional activity [34]. GSK-3β contributes to mitochondrial p53 apoptotic signalling as inhibition of GSK-3β by lithium or 2-Thio(3-iodobenzyl)-5-(1-pyridyl)-[1,3,4]-oxadiazole blocked cytochrome c release and caspase-3 activation [35]. However, the GSK-3β/p53 onco-suppressive effect loses its importance in many cancers including pancreatic cancer, where up to 76% of PDAC patients have a p53 mutation inducing its loss of function [36].

An important onco-suppressive effect of GSK-3β is that constitutively active GSK-3β was found to induce Snail degradation leading to increased E-cadherin and inhibition of the epithelial to mesenchymal transition (EMT), the driving force behind invasion and metastasis [37]. Inhibition of GSK-3β by lithium results in the upregulation of Snail and downregulation of E-cadherin in vivo [37]. In contradiction with this data, GSK-3β inhibitors such as BIO, TWS119, and LiCl, decreased the expression of mesenchymal markers in different cancer cell lines with a mesenchymal phenotype [38]. Furthermore, pharmacological and molecular inhibitions of GSK-3β reduced the migration ability of cancer cells. This effect is most likely induced by decreasing cancer stem cell-related cell surface marker CD44^+^/24^−^ as cancer stemness is associated with EMT and migration/invasion [38].

In a different study, GSK-3β was found to down-regulate the c-Myc level by causing its proteasomal degradation, leading to onco-suppressive effects including increased apoptosis and decreased proliferation [39]. However, a study using PDAC cells infected with lentivirus containing Cas9 and guide RNA (gRNA) to c-Myc or scrambled were treated with GSK-3β inhibitor SB-732881-H, to test whether c-Myc inhibition will rescue PDAC cells from apoptosis induced by the GSK-3β inhibition, showed that in scrambled gRNA-infected cells, GSK-3β inhibition induced apoptosis. In contrast, targeted knockout of c-Myc in cells infected with c-Myc gRNA abrogated GSK-3β inhibition induced apoptosis [28]. Taken together, apoptosis induced by GSK-3β inhibition in PDAC tumors is in part mediated by c-Myc, and therefore, the proposed onco-suppressive effect of GSK-3β through inhibition of c-Myc is not relevant to PDAC.

Finally, Ben-Josef et al. showed in 2015 that the GSK-3β expression level is a strong prognosticator in PDAC, independent of other known factors such as tumor (T) stage, nodal status, surgical margins and CA19-9 [40]. More importantly, they found that patient survival was significantly increased in PDAC patients with a high level of GSK-3β compared to patients with a low level of GSK-3β. However, the study included a relatively small number of patients (163 patients) and did not associate survival with the activity of GSK-3β or localization (nuclear vs. cytoplasmic) [40]. The increased protein level of GSK-3β is not necessarily associated with the increased activity of GSK-3β, making the conclusions of the study not relevant to the real association between GSK-3β and survival of PDAC patients.

Figure 2 illustrates the proposed tumour suppressor pathways regulated by GSK-3β in pancreatic cancer.

The above-mentioned studies lead to conclude that most of the data showing the anti-oncogenic effect of GSK-3β is either weak, not relevant to pancreatic cancer, or overcome by other pro-cancer effects. Most of these studies were published between 2001 and 2010 when the pro-cancer role of GSK-3β was not well known. Data published in the last 15 years have demonstrated the oncogenic role of GSK-3β, making it a very promising target for treating cancer. This data is summarized in the next section.

## 5. GSK3-β Oncogenic Effect in Pancreatic Cancer

Differently from the proposed onco-suppressive roles of GSK-3β, strong data has been accumulating for many years supporting the oncogenic role of GSK-3β [2,3]. In pancreatic cancer, GSK-3β is involved in regulating cancer cell proliferation, resistance to apoptosis and autophagy, chemo-resistance, and EMT, leading to invasion and metastasis [1,2,3,41]. 

GSK-3β is a key regulator of the NF-κB transcription factor responsible for the expression of genes that regulate proliferation, apoptosis, inflammation, angiogenesis, invasion, and chemo-resistance in pancreatic cancer cells [42]. GSK-3β nuclear accumulation was found to significantly correlate with human pancreatic cancer de-differentiation and regulation of NF-κB. Inhibition of GSK-3β activity by AR-A014418 or by a dominant negative form of GSK-3β represses its nuclear accumulation via proteasomal degradation and arrests pancreatic tumour growth in vivo and decreases NF-κB-mediated pancreatic cancer cell survival and proliferation in established tumour xenografts [22]. During the last few years, we have gained more knowledge about the mechanism through which GSK-3β regulates NF-κB. Medunjanin et al. identified the NF-κB Essential Modifier (NEMO) as a GSK-3β substrate that is phosphorylated within the N-terminal domain at different serine residues [43]. NEMO activates IκB kinase (IKK) leading to phosphorylation of IκB and, subsequently, its proteasomal degradation, thus liberating NF-κB for nuclear translocation, enhancing transcriptional regulation [43]. GSK-3β phosphorylates NEMO leading to its stabilization and therefore, degradation of IκB and enhanced activity of NF-κB. Using NEMO mutants which could not be phosphorylated by GSK-3β, Medunjanin found that TNFα-induced NF-κB activation was hampered [44]. 

In addition to the GSK-3β up-regulation of the NF-κB pathway, which has a broad pro-cancer effect, GSK-3β is also involved in another major pro-cancer pathway in PDAC, namely the k-*ras* pathway, which is activated in 95% of PDAC patients. First, GSK-3β is a downstream target of the pancreatic cancer initiating and promoting K-*ras* pathway. Mutant/active K-*ras* enhances GSK-3β promoter activity and protein expression via the MAPK signalling, leading to increased and sustained expression of GSK-3β in PDAC cells [45].

Next, Ras-mutated tumours were also associated with increased activation of mitogenic PI3k signalling, and reduced PTEN phosphatase activity, which both drive tumour growth and survival [46]. Therefore, increasing GSK-3β expression by induction of Ras-driven MAPK results in increased cancer cell plasticity and increased EMT. Pharmacological and molecular inhibitions of GSK-3β in K-*ras*-mutant PDAC cancer models showed upregulation of c-Myc and β-catenin leading to apoptosis induction, and indicative treatment selection of only malignant pancreatic epithelial cells [21,39]. 

Importantly, Palanivel et al. showed that GSK-3β regulates the Leucine-zipper-like transcriptional regulator 1 (LZTR1)-dependent mechanism that controls the stability of Ras proteins and proliferation of pancreatic cancer cells. LZTR1 protein is a substrate adaptor for E3 ubiquitin ligases that promote the degradation of both wild type and mutant Ras proteins. The data shows that GSK-3β inhibits LZTR1, leading to a sustained level of K-*ras* and increased proliferation of the pancreatic cancer cells [47]. 

Furthermore, in a recent study, mice expressing the inducible nuclear-targeted GSK-3β transgene were crossed with K-ras^G12D^;Pdx1-Cre (KC) mice, leading to profound pancreatic cyst development and expansion of cytokeratin-19^+^ ductal cells as early as at the age of 4 weeks. This data showed that nuclear GSK-3β worked in cooperation with K-ras mutation to promote the reprogramming of pancreatic progenitors to the ductal lineage, resulting in loss of acinar cells and expansion of ductal cells with PDAC characteristics. This data suggests a role of GSK-3β in the early stage of the PDAC development and confirms the strong pro-cancer role of GSK-3β [48].

Depletion of GSK-3β sensitized PDAC cells to tumour necrosis factor (TNF)-related apoptosis-inducing ligand (TRAIL) in PDAC cells and in PDAC animal models, and the inhibition of GSK-3β function by AR-A014418 or LY2064827 in PDAC cells impaired the expression of the anti-apoptotic Bcl-xL and cIAP2 proteins [49,50]. 

In addition to the effect of GSK-3β on proliferation and inhibition of apoptosis, GSK-3β phosphorylates/inhibits Cyclase-Associated Protein 1 (CAP-1), leading to increased migration and invasion of pancreatic cancer cells [51]. CAP1 phosphorylated at S308/S310 regulatory site is associated with the increased activity of GSK-3β observed in pancreatic cancer. This suggests the inhibition of GSK-3β could effectively inhibit cancer cell invasion through regulation of CAP1.

The viability of therapeutic targeting of GSK-3β has been demonstrated via pharmacological inhibition and RNA interference to reduce cancer cell survival and proliferation and induce apoptosis in gastrointestinal and pancreatic cancer cells, glioblastoma, hematologic malignancies, osteo-sarcoma, gynaecologic and urogenital cancers, and lung cancers [52,53]. The inhibition of GSK-3β significantly reduced the proliferation and survival of cancer cells, sensitized them to gemcitabine and ionizing radiation, and attenuated their migration and invasion. These effects were associated with decreases in cyclin D1 expression and Rb phosphorylation. The inhibition of GSK-3β by RNA interference or by its specific inhibitor AR-A014418 also altered the subcellular localization of Rac1 and F-actin and the cellular microarchitecture, including lamellipodia. Coinciding with these changes were the reduced secretion of matrix metalloproteinase-2 (MMP-2) and decreased phosphorylation of focal adhesion kinase (FAK). The effects of GSK-3β inhibition on tumour invasion, susceptibility to gemcitabine, MMP-2 expression and FAK phosphorylation were observed in vitro as well as in tumour animal models [17,52].

Transcriptomic sub-types of pancreatic cancer were found to acquire sensitivity or tolerance to GSK-3β inhibitors. Loss of genes that drive endodermal lineage specification, *HNF4A* and *GATA6*, switch the metabolic profiles from classical to squamous, with GSK-3β as a key regulator of glycolysis. The inhibition of GSK-3β using TDZD-8 or Tideglusib resulted in selective sensitivity in the squamous subtype [54].

In addition to PDAC proliferation, apoptosis, and EMT/invasion, GSK-3β also regulates autophagy. Recently published data showed that pancreatic cancer cells treated with the anti-cancer drug salicylanilide derivative Niclosamide (Nic) have increased phosphorylation/inhibition of GSK-3β leading to inhibition of non-canonical hedgehog/Gli cascade via upregulation of Sufu and Gli3 as well as stimulation of autophagy and causing pancreatic cancer cell death [55]. Inhibited serine 9 p-GSK-3β mediates Sufu/Gli3 cascade, negatively regulating Hh/Gli1 cascade and positively regulating mTORC1 autophagy-mediated cell death [55]. This Hh-GLI impairment has also been demonstrated in preclinical models by a combination of the GSK-3β inhibitor lithium and gemcitabine dosing in pancreatic cancer [56]. Other PDAC GSK3β inhibitory treatments revolve around synergistic pharmacological interaction with Gemcitabine to treat tumour stages ranging from preclinical models to refractory, to advanced or metastatic [26,57,58]. This synergistic inhibitory treatment catalyses on the varied GSK-3β involvement in multiple signalling pathways to increase DNA damage and impair DNA repair, as well as alter the tumour microenvironment to reduce cell migration, cancer stemness, and metabolic pathways.

Inhibiting GSK-3β activity results in decreased cellular ATP-production and increased AMP/ATP ratio which contributes to AMP-activated protein kinase (AMPK) activation. These effects are coupled to increased LC-3B biosynthesis and p62 protein reduction indicating autophagy induction. Liver kinase B1 (LKB1) is involved in AMPK activation, suggesting the therapeutic use of GSK-3β inhibitor molecules, such as TDZD8, Tideglusib, TWS119, and peptide L803-mts, that promote LKB1 translocation to cytoplasm and enhanced LKB1 interaction to prevent resulting autophagy [59]. By inhibiting GSK-3β we can provide a therapeutic approach to controlling autophagy induction by manipulation of the LKB1-AMPK pathway to selectively treat tumours or metastatic lesions. 

In addition, deregulation of GSK-3β expression and activity promotes tumour suppressor activity via JNK, Rb, Notch, TFEB, and C-Myc signalling pathways [60]. All this data strongly supports the pro-tumour function of GSK-3β. Figure 3 recapitulates the pro-oncogenic pathways regulated by GSK-3β in pancreatic cancer. 

Based on the previous data, multiple laboratories have been working on developing GSK-3β inhibitors for treating cancer. There are multiple ongoing clinical trials testing GSK-3β inhibitors in cancer patients including four studies in PDAC patients [61,62]. Table 1 shows a list of GSK-3β inhibitors with their therapeutic activity, diseases tested on, and observed results. The table also includes a list of ongoing clinical trials testing GSK-3β inhibitors in cancers.

## 6. Conclusions

For many years, the role of GSK-3β in cancer cells as a pro-oncogenic or tumour-suppressor protein has been a much-disputed topic, which has stunted therapeutic targeting of this kinase. It is now clear that GSK3-β is a powerful pro-cancer protein and is a perfect candidate for treating PDAC and many other cancers. 

The review of the tumour-suppressive role of GSK-3β in cancer reveals that most of these effects are either not active in pancreatic cancer, such as in the case of p53 regulation, or overcome by other pro-oncogenic effects of GSK-3β, such as in the case of β-catenin, c-Myc, and Snail pathways. On the other hand, the pro-oncogenic effects of GSK-3β are supported by strong data and interact with critical up-regulated proteins in the cancer progression such as K-ras and NF-κB. GSK3-β is expressed at higher levels in tumour cells and metastatic lesions compared to normal tissue. Therapeutic inhibition of GSK-3β has been shown to reduce cancer cell proliferation and survival in different malignant neoplasms, with combination treatments suggesting a viable method to avoid treatment resistance. The ongoing trials using GSK-3β inhibitors for treating PDAC and other cancers will reveal more data about this treatment strategy. GSK-3β is a very relevant target for treating PDAC. 

## Figures and Tables

**Figure 1 ijms-23-14133-f001:**
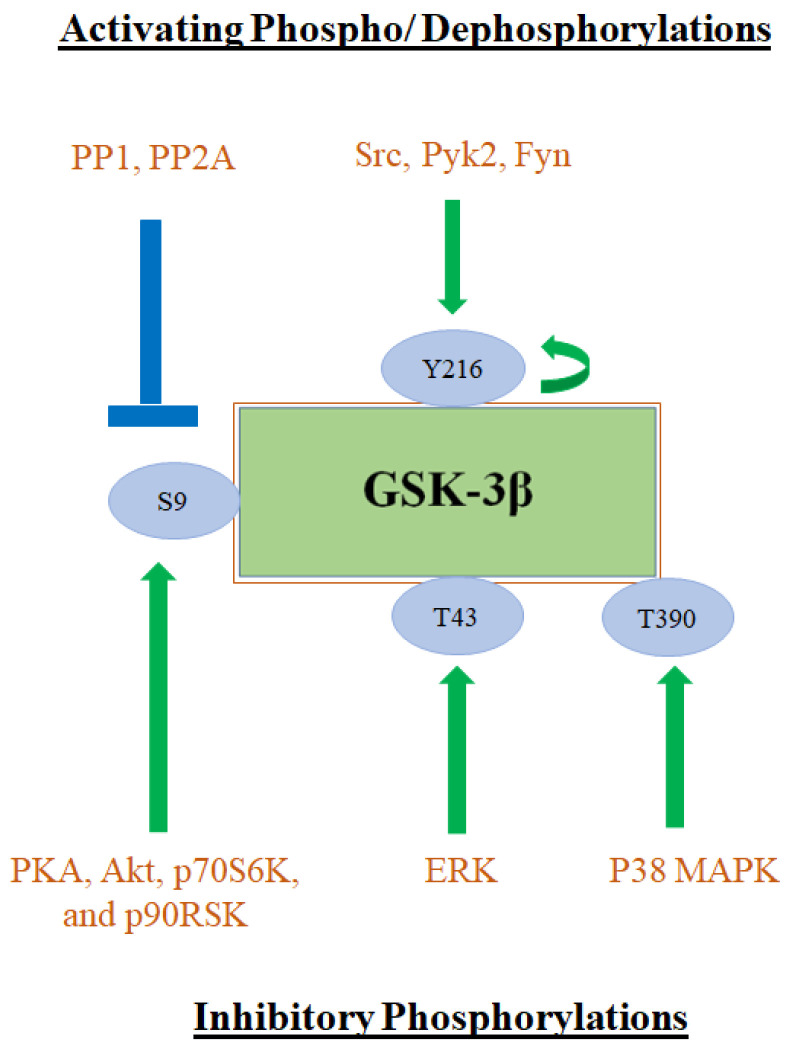
Phosphorylation sites of GSK-3β. Green arrows show phosphorylation. Blue lanes show dephosphorylation.

**Figure 2 ijms-23-14133-f002:**
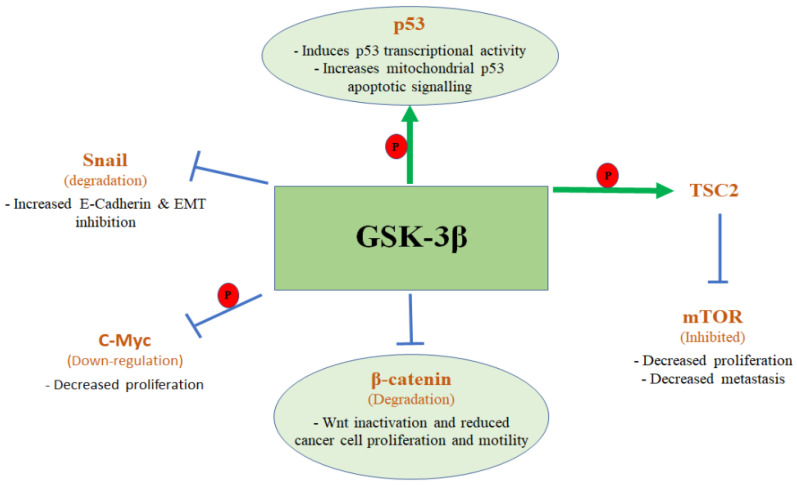
Proposed onco-suppressive pathways regulated by GSK-3β but found not relevant to pancreatic cancer.

**Figure 3 ijms-23-14133-f003:**
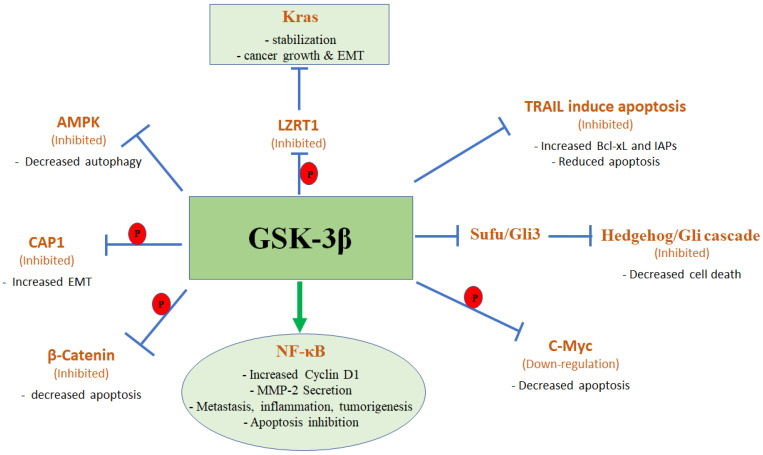
Oncogenic pathways up-regulated by GSK-3β.

**Table 1 ijms-23-14133-t001:** Inhibitors of GSK-3β in Cancer Treatment.

GSK-3β Inhibitors	Therapeutic Activity	GSK-3β Specificity	Resultant Targeting	Diseases Effected	Reference/Clinical Trial
benzofuran-3-yl- (indol-3-yl) maleimides	ATP Competitive	Yes	Supresses proliferation and survival of cancer cells	Pancreatic Cancer	[63]
9-ING-41	ATP Competitive	Yes	Impaired glucose metabolism Autophagy Differentiation of Renal Cancer	Breast Cancer, Renal Cell Carcinoma	[64] NCT03678883, 1801 phase 1/2 study
CHIR98014	ATP Competitive	Yes	ROS generation and mitochondrial dysfunction	Lung Cancer	[65] NCT01287520, 2018
Library of synthetic (1,2,4-oxadiazole) topsentin analogs	ATP Competitive EMT markers SNAIL-1/2 and metalloproteinase-9	Yes	Apoptosis Induction	Pancreatic Cancer	[66]
Metavert	Dual Inhibitor- GSK-3β/ HDAC	Yes (with inhibition of HDAC)	Reduced EMT markers and cancer cell survival	Pancreatic Cancer	[26]
Lithium (Lithium Carbonate)	Non-ATP Competitive- Mg2+ competitor	No	Mood stabilizer	Osteosarcoma, Esophageal Cancer	[56]
Tideglusib NP031112 NP-12	Non- ATP Competitive	Yes	Beta-Catenin C-MYC	Pancreatic Cancer	[28,67]
AZD-1080	ATP Competitive	Yes	Beta-Catenin C-MYC	Pancreatic Cancer	[7]
LY2090314	ATP Competitive-Phosphorylation Inhibition	Yes	TAK1	Pancreatic Cancer, Leukemia	[68] NCT01632306, 2019 NCT01287520 NCT01214603
SB-732881-H	ATP Competitive	Both α and β isoforms of GSK-3	Apoptosis Induction	Pancreatic Cancer	[65]
AR-A014418	ATP Competitive	Yes	NOTCH1, Cell differentiation/ proliferation inhibition	Pancreatic Cancer, Prostate Cancer	[69,70,71]
BIO	ATP Competitive- Phosphorylation Inhibition Inhibits CDKs	Both α and β isoforms of GSK-3	Caspase 3 Caspase 9 Beta-Catenin C-MYC Cell proliferation ROS generation and mitochondrial dysfunction	Glioblastoma multiforme, Pancreatic Cancer, Osteosarcoma, Lung Cancer	[28,72]
6BIO	Androgen receptor and signalling downregulation	Both α and β isoforms of GSK-3	Androgen Receptor Signalling Induced autophagy Reduced metastasis	Prostate Cancer	[73]
SB216763	ALP expression	Both α and β isoforms of GSK-3	Androgen receptor activity, cell proliferation FAK	Prostate Cancer, Cervical Adenocarcinoma, Colon Adenocarcinoma, Osteosarcoma	[73,74,75]
TWS119	Induces neuronal and CD8(+) T cell differentiation	Yes	γδT cell survival and proliferation activated	Breast Cancer, Colon Cancer, Lung cancer	[76]
SB415286	PD1	Both α and β isoforms of GSK-3	Improves osteoblastic differentiation	Lymphoma, Melanoma	[73,77,78]
ABC1183	CDK9 Beta Catenin TNFalpha/IL6	No	Growth inhibitory	Melanoma, Gastrointestinal cancers	[19,73]
TDZD-8	ATP non-competitive XIAP	Yes	Apoptosis	Neuroblastoma, Glioblastoma	[73,79]
CHIR99021	Wnt activator	Both α and β isoforms of GSK-3	Microtubule stability, Impaired chromosomal alignment	Non-small-cell Carcinoma	[65,73] NCT01287520, 2018
Indirubin (Indirubin-3′-oxime)	Inhibits CDKs Inhibits protein kinases	No	Cell cycle arrest and apoptosis	Leukemia	[73]
NP00111	ATP non-competitive	No	Protection against oxygen and glucose deprivation	Cancer	[80]
3F8	GD2	Yes	Enhanced cellular cytotoxicity	Neuroblastoma, Melanoma, Osteosarcoma, Medulloblastoma	[81,82,83,84,85,86]
Kenpaullone	Inhibits CDKs	No	Induced cytotoxicity and chemosensitivity	Glioblastoma, Breast Cancer	[87,88,89]
GS87	MAPK signalling activation	Both α and β isoforms of GSK-3	differentiation and growth arrest	Acute Myeloid Leukemia	[90]
2-Thio(3-iodobenzyl)-5-(1-pyridyl)-[1,3,4]-oxadiazole	Caspase-3 and cytochrome c inhibition	Yes	Mitochondrial p53 apoptotic signalling	Neuroblastoma and lung carcinoma	[35]
LY2064827	NF-*κ*B and TRAIL- or TNF*α* induced apoptosis	Both α and β isoforms of GSK-3	Induced chemosensitivity	Pancreatic cancer	[50]

## Data Availability

Not applicable.

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
