# Peer review of "Glycogen Synthase Kinase 3β: A True Foe in Pancreatic Cancer"

_ijms, 2022, doi:10.3390/ijms232214133_

Round 1

Reviewer 1 Report

Overall, an excellent review listing both sides of the argument and showing good experimental and clinical evidence of the oncogenic potential off GSK-3B and therefore its validity as a therapeutic target in PDAC.

I have only minor adjustments to request:

- Figure 1 could be made clearer with better arrows/symbols. As it is, despite the text, it looks like PP1/PP2A inhibit GSK-3B and the rest of the molecules activate it. Maybe use different colours too?

- Lines 60-63 should be rephrased as the message is not clear. It states the review will mostly discuss the anti-cancer roles of GSK-3B, when then you argue for its oncogenic role. Also, it says  'We do not discuss detailed pathways regulated by GSK-3β in cancer in general'. Please rephrase this and make it clearer.

- Figure 3 does not include the B-catenin pathway that was stated to be relevant for PDAC in lines 116-117. 

- some grammatical errors in line 58 ('looks', instead of 'look'), line 110 (there is an extra point), line 136 ('induing', instead of 'inducing'), line 296 ('that's', instead of 'which has'. This whole sentence is quite long and complex and I had to read it a couple times to understand it, so rephrasing is recommended).

Reviewer 2 Report

            The authors describe some the early concepts that link GSK-3b to b-catenin stabilization and cell growth and development but then go on to argue against the apparent misconception of GSK-3b as a suppressor of tumor development.  Part of the misconception stems from the established role of APC mutations and the stabilization of b-catenin as a driver for colon cancer. The authors mention that GSK-3b expression and nuclear localization can be elevated in cancers, as is the case for some other protein kinases, but these characteristics alone do not distinguish GSK-3b as an important contributor of cancer. However the authors describe numerous studies in which inhibitors of GSK-3b can lead to growth reductions or apoptosis in pancreatic cancer cells. An important disclosure is that the authors have an association with the development with one of these inhibitors (metavert) but in fairness they do not focus on this inhibitor alone. This inhibitor information supports the role of GSK-3b in cancer cell growth but not necessarily a role in cancer development. The analysis of inhibitors alone is not sufficient to claim an important role for GSK-3b, or any other protein, in the growth of cancers because inhibitors can have off-target effects that could play roles in the fate of cancer cells. The authors do mention some studies that also analyze the effects of GSK-3b depletion through RNA silencing which greatly strengthens support for a specific role of GSK-3b in cancer.

            The substrates of GSK-3b appear to be wide spread given the potential motifs that can be recognized by this kinase and the actions of GSK-3b are often guided by other protein kinases that provide priming phosphorylations in target substrates. Therefore it is not surprising, at least to this reviewer, that inhibition of GSK-3b would have consequences in the growth in cancer cells. This review provides a helpful reminder that many regulatory proteins such as GSK-3b can impact many different pathways and can have a variety of functions in different settings (tissue type, disease, developmental stage). This review will be of particular interest for those studying pancreatic cancers.

Concerns –

The authors should explain if any of the inhibitors they use in their argument have been analyzed for GSK-3b specificity and also indicate that inhibitor specificity is an assumption for supporting their argument. Inhibitors of protein kinases can potentially have cross reactivity with other kinases. In addition the authors should indicate which cases of GSK-3b inhibition are backed up with GSK-3b depletion analysis.

The authors mention many studies that involve GSK-3b inhibitors but do not specify which inhibitor. It would be useful to indicate which specific GSK-3b inhibitor is being referred to in their text describing these studies because inhibitors can have very different specificities and efficacies.

Minor

Several sentences need referencing (e.g., lines 113, 153,188, 202 and probably many more). The reference should be included at the end of the first sentence that mentions a particular study rather than at the end of multiple sentences that refer to the same study.

Ref’s 27 and 59 are the same (Edderkaoui et al.)

Line. 152 GSK-3b (inhibitor?), to test . . .

Line 175 “Strong” is the description appropriate?

Line179 – small paragraph can be deleted as it is a conceptual repeat of the above paragraph

Line 181 – unusual to have a one sentence paragraph
